# Government responses and COVID-19 deaths: Global evidence across multiple pandemic waves

**Thomas Hale**[1]*, **Noam Angrist**[1], **Andrew J. Hale**[2], **Beatriz Kira**[1‡], **Saptarshi Majumdar**[1], **Anna Petherick**[1‡], **Toby Phillips**[1‡], **Devi Sridhar**[3‡], **Robin N. Thompson**[4,5‡], **Samuel Webster**[6‡], **Yuxi Zhang**[1]

**1** Blavatnik School of Government, University of Oxford, Oxford, United Kingdom, **2** Larner College of Medicine at the University of Vermont, Burlington, Vermont, United States of America, **3** Professor, University of Edinburgh, Edinburgh, Scotland, United Kingdom, **4** Mathematics Institute, University of Warwick, Coventry, United Kingdom, **5** Zeeman Institute for Systems Biology and Infectious Disease Epidemiology Research, University of Warwick, Coventry, United Kingdom, **6** Unaffiliated, London, United Kingdom

☯ These authors contributed equally to this work.
‡ These authors also contributed equally to this work.
* Thomas.hale@bsg.ox.ac.uk

**Data Availability Statement:** All underlying data are freely available, and continuously updated, on the website of the Oxford COVID-19 Government Response Tracker: https://www.bsg.ox.ac.uk/

## Abstract

We provide an assessment of the impact of government closure and containment measures on deaths from COVID-19 across sequential waves of the COVID-19 pandemic globally. Daily data was collected on a range of containment and closure policies for 186 countries from January 1, 2020 until March 11[th], 2021. These data were combined into an aggregate stringency index (SI) score for each country on each day (range: 0–100). Countries were divided into successive waves via a mathematical algorithm to identify peaks and troughs of disease. Within our period of analysis, 63 countries experienced at least one wave, 40 countries experienced two waves, and 10 countries saw three waves, as defined by our approach. Within each wave, regression was used to assess the relationship between the strength of government stringency and subsequent deaths related to COVID-19 with a number of controls for time and country-specific demographic, health system, and economic characteristics. Across the full period of our analysis and 113 countries, an increase of 10 points on the SI was linked to 6 percentage points (P < 0.001, 95% CI = [5%, 7%]) lower average daily deaths. In the first wave, in countries that ultimately experiences 3 waves of the pandemic to date, ten additional points on the SI resulted in lower average daily deaths by 21 percentage points (P < .001, 95% CI = [8%, 16%]). This effect was sustained in the third wave with reductions in deaths of 28 percentage points (P < .001, 95% CI = [13%, 21%]). Moreover, interaction effects show that government policies were effective in reducing deaths in all waves in all groups of countries. These findings highlight the enduring importance of non-pharmaceutical responses to COVID-19 over time.

research/research-projects/coronavirus-government-response-tracker.

**Funding:** The authors received no specific funding for this work.

**Competing interests:** The authors have declared that no competing interests exist.

## Introduction

The COVID-19 pandemic has upended healthcare, cultural, financial, and government systems worldwide. While vaccines are being deployed rapidly one year after the outbreak began, control of the COVID-19 pandemic continues to rely largely on government non-pharmaceutical interventions (NPIs) [1–6]. Such interventions have proliferated worldwide, including school closings, travel restrictions, public gathering bans, and stay-at-home orders [7]. These policies aim to create physical distancing or otherwise slow the spread of COVID-19, often in concert with testing and contact tracing regimes of varying robustness [3, 8–11]. In some cases, closure and containment measures have been extreme, with unprecedented social, cultural, and financial implications [12–14]. Governments have varied significantly in both the degree of their interventions and how quickly they adopt them [15, 16].

Robust evidence now shows that, under most conditions, early adoption of stringent NPIs is associated with a reduction in transmission [16–27]. However, with full vaccination still elusive, and with the possibility of vaccine-escaping variants emerging, NPIs can be expected to continue playing a significant role in managing the pandemic in the foreseeable future [28–31]. A critical question is therefore whether we can expect NPIs to continue working as the pandemic stretches into longer time frames. On the one hand, we might expect effectiveness to improve over time as governments learn how to better calibrate and target policies [32, 33]. On the other hand, we may expect the opposite to the extent individuals grow tired of COVID-19 restrictions, as responses become politicized, or as economic disruption grows more severe [34–39].

As a first step toward investigating these trends, we explore the average effects of NPIs on the spread of the disease globally. We present a global analysis of governments' responses to date, and a global assessment of their relationship to the spread of the pandemic over time. We tracked 186 governments' responses across a series of non-pharmaceutical interventions and created a composite index that captured how, over time, each country's government responded. Looking at peaks and troughs in countries' death tolls, we divide each outbreak into successive waves, limiting our analysis to the 113 countries that recorded at least one death per day, on average, in the period of analysis. We hypothesized that the overall stringency of governments' interventions would affect the subsequent rate of deaths related to COVID-19 in each wave, comparing the effects of policies and deaths both across countries and across waves. This article seeks to make three contributions. First, we explore whether the effects of policies on deaths varies across each wave of the pandemic. Second, we develop an approach to operationalize in observational analysis an approach to variation in the effect of policy over time, a feature of some epidemiological models [40]. Third, our analysis includes a global distribution of countries experiencing significant outbreaks of COVID-19.

## Methods

### Data collection

We collected information on 186 national governments' responses across a range of NPIs (see S1 Table in S1 File). These measures were recorded for each day in each country, creating a measure of variation in government responses both across countries and across time. Data were collected by the authors and trained research assistants from publicly available sources such as news articles, government briefings, and international organizations. Data collectors coded government responses on a simple binary or ordinal scale registering the stringency of a given policy. Indicators C1-C7 and H1 (see S1 Table in S1 File) were further classified as either "targeted" (meaning they apply only in a geographically concentrated area) or "general"

(meaning they apply throughout the entire jurisdiction). The data cover over 186 countries from January 1, 2020 onwards, though we limit our analysis up until March 11[th], 2021 and to only the 113 countries that experienced one or more deaths per day, on average, during the period of analysis. Because we do not use human data or tissue, or involve human subjects, approval by the university review board was not required. To ensure accuracy and consistency, data collectors were required to pass an online training and to participate in regular team review meetings. Each data point was verified by at least two data collectors independently, and includes notes and source materials to substantiate each observation. Importantly, for all NPIs, we record only the official policies at the national level, not how well they are implemented or enforced.

## Measuring government non-pharmaceutical interventions

Our primary measure of governments' NPIs is a composite Stringency Index (SI) that records the number and restrictiveness of government containment and closure measures, calculated as follows. For each of the nine relevant policy response indicators, we create a score by taking the ordinal value, adding 0.5 if the policy is general rather than targeted, and rescaling each of these to range from 0–100. Conservatively, we assign a score of 0 to any indicators missing data, and reject any country-days where more than one of the indicators is missing. The mean of these nine scores gives the composite SI.

We rely chiefly on this simple, unweighted SI because this approach is most transparent and easiest to interpret [41]. Composite indices have the value of facilitating comparison across countries, albeit with the trade-off of condensing information. In practice, however, we observe that most countries in the time period of analysis adopted most NPIs as a package, further justifying the use of a composite measure [7]. The robustness of SI is also supported by the high correlations between this and alternative indexes in both level of stringency and shifts over time [42].

The degree of government response is measured by the value of the SI for a country on a particular day, or, in the cross-sectional analyses, the average level of stringency from January 1, 2020 to March 11[th], 2021.

In addition to the nine indicators that comprise the stringency index, we include as a control measures of governments' testing and contact tracing (S1 Table in S1 File). This controls for the potential that effects on deaths are confounded with increased identification of COVID-19 cases and deaths. These are also recorded on an ordinal scale representing the breadth and thoroughness of the policy. Additionally, we use an additional index, the government response index, as a robustness measure. The government response index consolidates an expanded set of indicators, including, for example, economic support and vaccination policies.

## Dividing countries into outbreak weaves

We identify each country's respective outbreak "waves." Accounting for specific waves is critical. First, this is important to answer the question of whether the effectiveness of government policies is sustained over time and across different phases of the pandemic. Second, accounting for waves has a methodological advantage of enabling more precise calibration to the specific slope of the curve in each wave, rather than applying a one-size-fits-all estimation technique across all waves. Even within a country, waves often have substantially different peaks, troughs, and slopes.

Studies on COVID-19 and the previous ten virus-driven pandemics since 1889 have found similar wave patterns [43, 44]. There is no scientific agreement on the definition of a "wave",

or on the question of whether the driving force of this pattern is human behaviour or the spread of the virus [45, 46]. Yet, the literature includes various models to predict future waves, and some studies elaborate different time-variant associations between the spread of the virus and non-epidemic dynamics such as human behaviour, government measures and enforcement, as well as environmental factors, such as temperature [33, 47–51]. Some studies find COVID-19 waves can share characteristics, such as the vulnerability of the elderly population; others find the demographic structure of cases and deaths shift substantially across waves [52, 53]. Researchers have also found that different waves in a single country may be dominated by clusters and outbreaks in dispersed locations [54].

While the literature does not offer a precise definition of "waves", there is a broad consensus that a wave is a phase of disease that is more substantial than a "sporadic outbreak", and that comprises a rising phase and a subsequent falling phase [46, 55]. As a first step in identifying waves, we therefore locate peaks and troughs for each wave in each country and split waves between troughs accordingly. We smooth curves using a local non-parametric polynomial regression, a generalized form of a moving average, to avoid misidentification of waves due to noisy data or fluctuation in reporting of deaths. Specifically, we use locally weighted scatterplot smoothing, also known as LOWESS regression. We identify waves as the point before and after troughs. We consider waves distinct if their troughs are more than approximately one month apart, otherwise we include this together as one wave. We chose this threshold to correspond to the lag we hypothesize between NPIs and deaths, discussed below. In addition, we only count a wave if more than 20 cases occurred in that given wave (to distinguish a wave from a sporadic outbreak). Given the lack of consensus in the literature on how to define a wave, we believe the approach selected is transparent and simple, facilitating analysis, but we do not claim it is the only or best way to divide the phases of the pandemic. Finally, we consider the period from early January 2020 to March 11[th], 2021. Since then, additional waves have emerged in multiple countries, however, these waves are ongoing and the effects of policies on deaths during these waves are not yet able to be fully detected.

For example, Fig 1 below shows peaks and troughs in India, South Africa, and the United States and. As of March 2021, India has one major and extended wave from April 2021-January 2022; South Africa had 2 waves: the first in July-September, and the second in December-February; and the United States had 3 waves: the first in April-May, the second in July-August, and the third in December-February. These examples clearly highlight the need to take a data-driven approach to identifying waves. Waves occur in different countries at different points in time, and last for different lengths of time. Moreover, a standardized approach is needed to draw boundaries between waves and sporadic outbreaks or "noisy" data. Our approach is certainly not the only way to identify waves, but provides a useful tool to facilitate analysis of the effect of NPIs.

## Outcome variables

We estimate the relationship between government interventions and the intensity of the COVID-19 outbreak by country. Information on confirmed COVID-19 cases and deaths were taken from John Hopkins University [56]. However, the true number of cases and deaths, as well as the reproduction number, are the subjects of significant uncertainty and a major topic of ongoing research [12, 57–59]. Observationally, the true number of cases is difficult to measure consistently because different countries have tested for COVID-19 more or less widely, and report case information in variable ways [3, 60]. We take a conservative approach and use recorded deaths as our main outcome, which we expect to be reported more consistently and captures the public health consequence of the epidemic most directly.

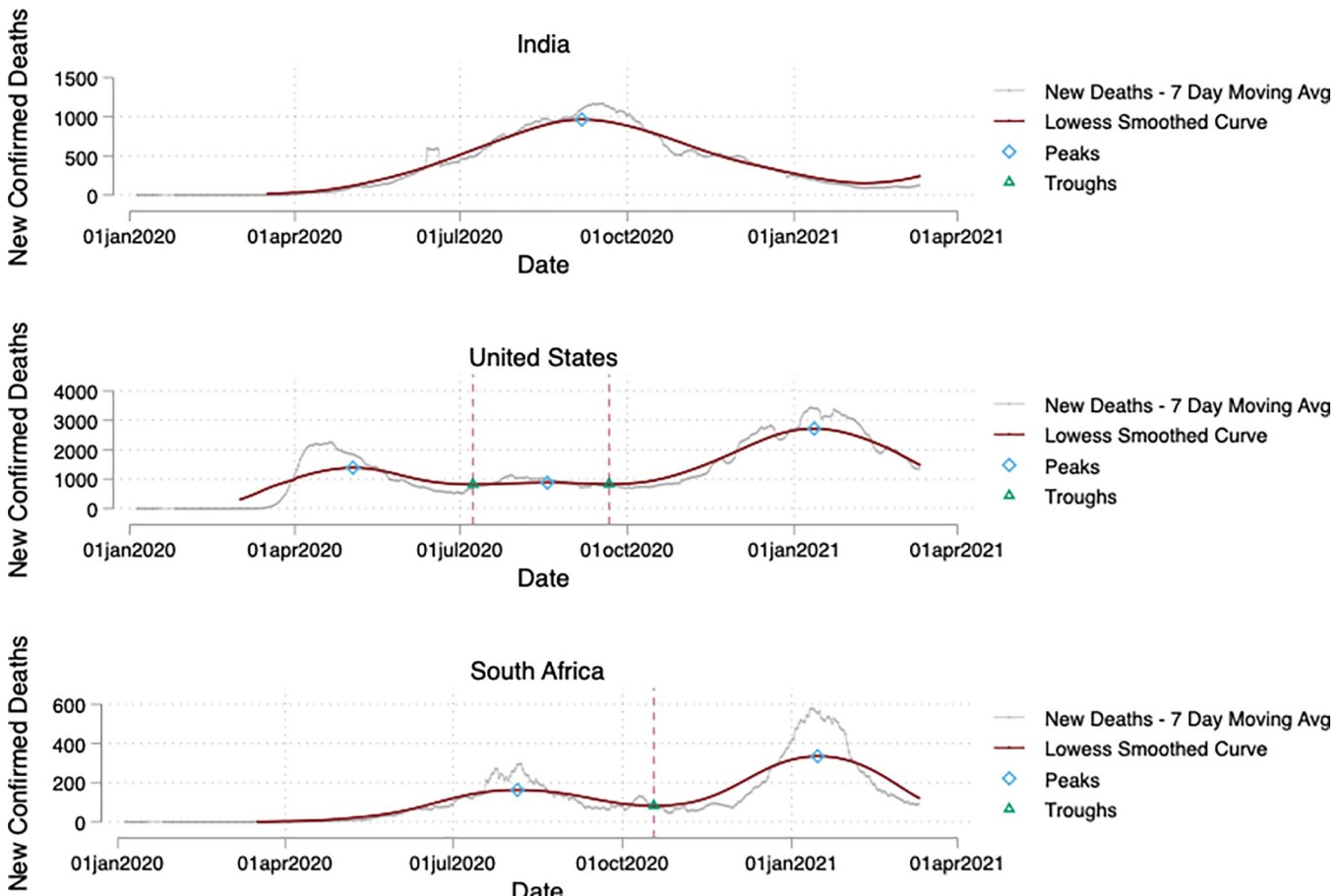

**Fig 1. Identifying waves of COVID-19 in different countries.** India, South Africa, and the United States exemplify the need for data-driven distinctions for enumerating waves, which occurred in different months, to different degrees and over different time horizons.

We analyse deaths in terms of the daily natural log in deaths in a country. We use the log transformation to account for the exponential trajectory of the epidemic growth curve. This specification also facilitates analysis because the first differences in logs approximate percentage changes, meaning the coefficients from our regressions are readily interpretable.

## Statistical analysis

Our approach aligns with well-established empirical methods to study how government measures address historical epidemics and during the COVID-19 pandemic [61, 62]. We estimate how the degree of government response relates to changes in deaths using Ordinary Least Squares (OLS) regression, with the country-day as the unit of analysis. We include country fixed effects to control for variations in country-specific factors. To this end, effects of policies are estimated based on *within* country changes in deaths for 113 countries [63]. Importantly, country fixed effects control for all country-specific characteristics that do not vary over the period of analysis, such as the level of wealth, pre-existing robustness of the health care system, the government's overall capacity to implement policy, or the population's general tendency to follow government advice or not. We also control for deaths at the beginning of the time period of analysis to account for variations in baseline deaths in a given country and wave. As a robustness check to ensure county fixed effects models are not biased, we also estimate

models that directly include controls for country-specific factors using data from the World Bank; these controls include adult mortality rates, hospital bed per 1000 people, number of physicians per 1000 people, prevalence of diabetes, population density, total population, and GDP per capita. Additional robustness checks use subnational jurisdictions in countries with substantial policy heterogeneity, such as the United States and Brazil. We also include robustness tests with an alternative index, the Government Response Index, as well as including controls for testing and contact tracing policies.

We estimate these models across all country-days during the entire period of analysis from January 2020 to March 11th 2021. We include countries with at least one death per day on average over this period. Since effects are identified based on *within* country changes in deaths, countries that had very few deaths will not have enough variation to enable a credible analysis. If we did not include country fixed effects, identification would also be derived from across-country differences in policy and resulting deaths. However, we avoid this estimation approach since it introduces bias in the form of confounded variables such as wealth, age-structure, and health care capacity variation across countries. 113 countries are included in the final sample. We estimate wave-specific effects using interaction terms for each wave, interacted with the stringency index.

This kind of observational study faces two key inferential challenges: how to control for the effect of time, including the "natural" growth and diminishing of the disease, which is unobserved, as well as the possibility of reverse causality or other forms of endogeneity. For example, a positive relationship between current stringency and new deaths does not necessarily mean stringency increases deaths; rather deaths might trigger a policy response. We address these issues in three ways.

First, we lag the explanatory variable by four weeks. This period reflects our best estimate, based on the existing literature, of the lag between behavioural change, transmission, the emergence of disease, and, ultimately, death. Studies find that NPIs significantly reduces the reproduction number of SARS-CoV-2 in the subsequent 28 days after introducing the policies, provided public events are banned throughout [64]. However, the event of introducing a lockdown contributes very little to mitigating a pandemic wave after 30 days [65]. Second, we use log of deaths as the dependent variable, approximating the exponential growth rate of the epidemiological curve. Third, we include a time trend in regressions to account for natural growth patterns in deaths as well as normalize the time period to account for the time in which the epidemic reached different countries across the globe. With this specification, which is now common in the literature, we identify likely effects of NPIs on the number of deaths from COVID-19. However, we suggest a modest interpretation of causality given the inherent challenges of identification for this research design.

## Results

### Variation in the speed and intensity of government responses and waves of disease

We observed significant variation in both the level of stringency and the time at which policies are adopted across national governments. While there is a near-universal increase in countries adopting containment and closure measures over time, with most countries moving to stringent measures after the first week of March 2020, the varied spread of the disease globally means that some countries adopted "lockdown" measures before local transmission began and some after. As the pandemic has progressed, countries increasingly differ in the NPIs they adopt. While information campaigns, international travel controls, and testing and contact measures tend to remain in place, closure and containment policies have waxed and waned[7].

We also observe variation in how many waves of disease countries experience. At the time of writing, most countries (63) have experienced two waves, 40 have only experienced one and 10 have experienced three or more. The number of waves continues to evolve, revealing the need for a wave-specific analysis and motivating future work on this subject. In the Supplement, we include figures for a series of example country and their respective waves.

## Regression results

Our core question is how NPIs, as measured by the Stringency Index, affect deaths in each country during each wave. Table 1 presents the results. Each coefficient shows how a one point increase in the stringency of government response four weeks prior relates to the difference in log of daily deaths in a given country on an average day. Further models are reported in the supplement as robustness checks (S2.1–S2.4 Tables in S1 File). Overall, the results strongly indicate that more stringent responses led to fewer deaths. Column (1) shows that 1 point of higher stringency 28 days in the past leads to an average reduction in deaths in all countries across all waves of -.006 (P < 0.001, 95% CI = [-0.005, -0.007]). In countries with one wave (column 2), we find an estimated reduction of -.005 (P < 0.001, 95% CI = [-0.003, -0.007]) with each additional point of stringency. In countries with a second wave (column 3), we find a reduction in deaths of -.004 (P < 0.001, 95% CI = [-0.002, -0.006]) in the first wave and -.008 (P < 0.001, 95% CI = [-0.006, -0.01]) in the second wave. In countries with a third wave (column 4), we find a reduction of -.021 (P < 0.001, 95% CI = [-0.017, -0.025]) in the first wave, -.031 (P < 0.001, 95% CI = [-0.027, -0.035]) in the second wave and -.028 (P < 0.001, 95% CI = [-0.024, -0.032]) in the third wave. All p-values in all cases are P < .001 indicating highly statistically significant results. Since first differences in logs approximate percentage changes and are asymptotically identical at small values, these effects can be interpreted as percentage

**Table 1. Association of government response stringency and deaths by wave.**

| | (1) | (2) | (3) | (4) |
|---|---|---|---|---|
| | Pooled Estimates for All Countries | One-Wave Countries | Two-Wave Countries | Three-Wave Countries |
| **LAGGED STRINGENCY BY 28 DAYS** | -0.006*** | | | |
| | (0.001) | | | |
| | [0.000] | | | |
| **LAGGED STRINGENCY BY 28 DAYS: WAVE 1** | | -0.005*** | -0.004*** | -0.021*** |
| | | (0.001) | (0.001) | (0.002) |
| | | [0.000] | [0.000] | [0.000] |
| **LAGGED STRINGENCY BY 28 DAYS: WAVE 2** | | | -0.008*** | -0.031*** |
| | | | (0.001) | (0.002) |
| | | | [0.000] | [0.000] |
| **LAGGED STRINGENCY BY 28 DAYS: WAVE 3** | | | | -0.028*** |
| | | | | (0.002) |
| | | | | [0.000] |
| **COUNTRIES** | 113 | 40 | 63 | 10 |
| **R2** | 0.76 | 0.75 | 0.76 | 0.81 |
| **COUNTRY FIXED EFFECTS** | Yes | Yes | Yes | Yes |
| **TIME TREND** | Yes | Yes | Yes | Yes |
| **LAGGED DEATHS CONTROL** | Yes | Yes | Yes | Yes |

*Notes*: All regressions coefficients are included in the table, followed by standard errors in parentheses and p-values in square brackets. Stars signify statistical significance at conventional thresholds.

points. This means that a 10-point difference in SI would be expected to lead, four weeks later, to 28 percentage point fewer deaths during the third wave. We further note a high R-squared of .75 to .81 across all regressions, indicating that our estimation model fits the data well. Because different groups of countries experienced one, two, or three waves of disease, it is not possible to compare the magnitudes of the coefficients across all columns in Table 1. However, the overall results clearly show an enduring relationship between policies and deaths across countries and across waves.

We also include a series of robustness tests in the S2.1–S2.4 Tables in S1 File. These robustness tests show analysis using subnational units for large heterogeneous countries such as the United States and Brazil. Additional robustness tests include specific controls, such as GDP hospital bed availability and comorbidities, testing and contract tracing policies, as well as using an alternative index, the government response index. All robustness tests show similar trends, with consistent relationship between more stringent NPIs and fewer deaths, and persistent effects across multiple waves of the pandemic.

## Discussion

Our data show that government responses do indeed have a statistically robust and substantively significant relationship with deaths related to COVID-19. Moreover, this relationship endures across multiple waves of disease. At the same time, the findings reveals that the effect of NPIs has varied over time and across countries to some degree. This might be for a variety of reasons: for example, perhaps vaccine rollouts coupled with government policy counteracted potential fatigue in adherence with policies, resulting in sustained effectiveness of government policies in reducing deaths. In this paper our aim is not to identify precise reasons for differences in effectiveness across waves, which likely diverge across countries, but rather to identify the first-order patterns in the effectiveness of NPIs over time. This motivates future research into effectiveness of policies during specific waves of the pandemic.

Our study has several limitations. Like any policy intervention, the effect of the responses we measured is likely to be highly contingent on local political and social contexts. For instance, the state-by-state level response in the United States has been heterogeneous, while our primary analysis is conducted at the national level (however, as S2.3 Table in S1 File shows, these patterns also hold across subnational jurisdictions). Nor do we measure the extent to which government interventions are successfully implemented. In addition, the relationships reported do not account for potential confounders that might have otherwise reduced deaths, such as seasonality, climate, or spontaneous behavioural change in response to changing risk perceptions or social norms in the population. While these factors have not yet been fully established for COVID-19, if they are, they will need to be accounted for to more reliably estimate the effect of government policies on growth in deaths. In spite of these limitations, our approach offers a global and comprehensive view of governmental response to COVID-19 to date with the best information available. By measuring a range of indicators, composite indices mitigate the possibility that any one indicator may be over- or mis-interpreted. By the same token, composite measures also make strong assumptions about what kinds of information are included. If the information left out is systematically correlated with the outcomes of interest, or systematically under- or overvalued compared to other indicators, such composite indices may introduce measurement bias.

Our results are in line with findings of changes in NPIs and the development of pandemics over time. Multi-country analyses have found the implementation NPIs was significantly associated with an overall reduction in COVID-19 incidence [61]. An analysis of the effect of five policy responses on COVID-19 deaths in 11 European countries found a significant impact of

interventions implemented several weeks before late March 2020, though these results were strongly driven by the experiences of Spain and Italy [16]. These findings hold in single-country studies and those focusing on subnational regions. A modelling study finds that without NPIs the number of COVID-19 cases would have been 51 to 125 folds higher in different cities and provinces in China [66]. The NPIs implemented in New York City were estimated to have reduced cases numbers by 72% and deaths by 76% [67]. While some studies shows the relationship between single NPI and the reduction in COVID-19 incidences or reproduction number, others try to identify the more effective blends of NPI sub-categories [68–70]. However, researchers also find the synergistic benefits of implementing a suite of multiple NPIs, which lends support to our approach of assessing the relationship between overall NPIs and deaths [71, 72].

Apart from the general association between NPIs and the spread of disease, researchers have also looked at how such relationships change over the course of previous pandemics. For example, earlier stringency measures decreased death rates in the 1918–1919 influenza pandemic by as much as 50% [73, 74]. That pandemic was characterised by three waves, with second and third waves occurring only after the relaxation of the "main battery of NPIs" [43, 75]. More recently, researchers start to look at how the association between NPIs and cases and deaths vary across different COVID-19 waves. A study investigating the performance of different NPIs across waves in 133 countries finds the most effective NPI blends change from gathering restrictions, facial coverings and school closures to facial coverings, gathering restrictions and international travel restrictions in the second wave. It concludes that the impact of NPIs had obvious spatiotemporal variations across countries by waves before vaccine rollouts [76]. According to a studying on 114 subnational areas in 7 European countries, the combined effectiveness of 17 NPIs on reducing local cases and deaths is still statistically significant, but the effect sizes in the second wave were smaller relative to that in the first wave [33]. The global analysis of varying association between NPIs and COVID-19 deaths in our paper echoes these findings and contributes to the emerging evidence base on how the relationship between NPIs and the spread of COVID-19 varies or remains constant over time.

Going forward, it will be important to continue monitoring government responses as the pandemic evolves. More granular analyses looking at the implementation and effectiveness of national policies, the role of individual measures and various combinations of policies, as well as the role of subnational governments or other social institutions can contribute further to this line of investigation. Further work should seek to explain variation in the effectiveness of NPIs across countries, examining both cross-sectional and longitudinal factors. Of particular importance, researchers can examine the role of path dependency, assessing how experiences in earlier waves affect the effectiveness of NPIs in later waves.

## Supporting information

**S1 File.**
(DOCX)

## Acknowledgments

We are grateful to the strong support from students and staff at the Blavatnik School of Government and across Oxford University for contributing time and energy to data collection. We thank Rafael Goldszmidt, Andy Eggers, and Devika Singh for helpful comments.

## Author Contributions

**Conceptualization:** Thomas Hale, Andrew J. Hale, Beatriz Kira, Anna Petherick, Toby Phillips, Devi Sridhar, Robin N. Thompson, Samuel Webster, Yuxi Zhang.

**Data curation:** Thomas Hale, Beatriz Kira, Saptarshi Majumdar, Anna Petherick, Toby Phillips, Samuel Webster, Yuxi Zhang.

**Formal analysis:** Thomas Hale, Noam Angrist, Saptarshi Majumdar, Anna Petherick.

**Investigation:** Thomas Hale, Noam Angrist, Beatriz Kira, Anna Petherick, Devi Sridhar.

**Methodology:** Thomas Hale, Noam Angrist, Andrew J. Hale, Anna Petherick, Robin N. Thompson.

**Project administration:** Thomas Hale, Beatriz Kira, Toby Phillips, Samuel Webster.

**Software:** Saptarshi Majumdar, Samuel Webster.

**Supervision:** Toby Phillips.

**Validation:** Andrew J. Hale.

**Visualization:** Saptarshi Majumdar, Toby Phillips.

**Writing – original draft:** Thomas Hale, Noam Angrist, Andrew J. Hale, Beatriz Kira, Anna Petherick, Toby Phillips, Yuxi Zhang.

**Writing – review & editing:** Thomas Hale, Noam Angrist, Andrew J. Hale, Beatriz Kira, Saptarshi Majumdar, Anna Petherick, Toby Phillips, Devi Sridhar, Robin N. Thompson, Yuxi Zhang.

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
