## [Decision Letter · Decision Letter 0]

12 Oct 2020

PONE-D-20-20672

Global assessment of the relationship between government response measures and COVID-19 deaths

PLOS ONE

Dear Dr. Hale,

Thank you for submitting your manuscript to PLOS ONE. After careful consideration, we feel that it has merit but does not fully meet PLOS ONE’s publication criteria as it currently stands. Therefore, we invite you to submit a revised version of the manuscript that addresses the points raised during the review process.

We look forward to receiving your revised manuscript.

Kind regards,

Holly Seale

Academic Editor

PLOS ONE

Journal Requirements:

2. PLOS ONE publication criteria and journal policy require methods and data to be described in enough detail to allow suitably skilled investigators to fully replicate and evaluate your study.

In addition, authors must make all data underlying the findings described in their manuscript fully available without restriction.

We note that you state "Data were collected by the authors and trained research assistants from publicly available sources such as news articles, government briefings, and international organizations."

However, the sources are not listed and data is not provided.

Please include a table of the primary data used for your study with the sources of the data listed in a supplementary file.

Reviewers' comments:

Reviewer's Responses to Questions

**Comments to the Author**

1. Is the manuscript technically sound, and do the data support the conclusions?

Reviewer #1: Partly

Reviewer #2: Partly

2. Has the statistical analysis been performed appropriately and rigorously? 

Reviewer #1: No

Reviewer #2: No

3. Have the authors made all data underlying the findings in their manuscript fully available?

Reviewer #1: No

Reviewer #2: No

4. Is the manuscript presented in an intelligible fashion and written in standard English?

Reviewer #1: No

Reviewer #2: Yes

5. Review Comments to the Author

Reviewer #1: Here is a list of specific comments. Note: line and page numbering in reviews and comments is based on ruler applied in Editorial Manager-generated PDF.

1. There was no page number and line number. It might be difficult to locate the following comments precisely.

2. Methods, Data Collection: The section needed more details. There were a lot of words without details. The reproducibility might be low.

3. Methods, Data Collection, “data were collected by . . . ”: I suggest describing the data source in more details to ensure reproducibility.

4. Methods, Data Collection, “data collectors coded govenment responses . . . ”: For the ordinal scale, did you mean C1 in Table S1? If so, please refer it to Table S1. In addition, please provide more details on the binary scale.

5. Methods, Data Collection, “several indicators were . . . ”: Similarly, please refer “several indicators” to Table S1.

6. Methods, Data Collection, “the data over over 170 countries . . . ”: Would it be possible to list all countries (exactly how many countries) in the supplementary materials and indicate which countries’ data contributed to which parts of analyses?

7. Methods, Measuring the speed and degree of government non-pharmaceutical interventions, “foreach of the nine relevant policy response indicators . . . ”: I believed the relevant policy response indicators were listed in Table S1. If so, please point them out and spend some sentences describing them.

8. Methods, Measuring the speed and degree of government non-pharmaceutical interventions, “most countries in the time period of analysis adopted most NPIs as a package”: I failed to understand this from Figure S1.

9. Methods, Measuring the speed and degree of government non-pharmaceutical interventions: Were the principal component analysis and principal factor analysis suitable for non-normal variables (i.e., ordinal and/or binary)?

10. Methods, Measuring the speed and degree of government non-pharmaceutical interventions, “the average level of stringency from January 1, 2020 to May 27, 2020”: Why the average start January 1, 2020, but not the date when the first case recorded for a country?

11. Methods, Outcome Variables, “we analyze growth in deaths in terms of the daily log difference . . . ”: Was the difference from that of the previous day? What if the difference was negative?

12. Methods, Outcome Variables, “we consider the maximum daily number of new deaths . . . ”: Would the maximum daily number of new deaths be a fair measurement because not all countries recorded the first death on the same date? Wouldn’t the countries recorded the first death on a later date within the study window have a relatively smaller maximum? Should the maximum be measured at a fixed window since the first recorded death?

13. Methods, Statistical Analysis, “we estimate both cross-sectional models ...”: Somewhere in the Methods section, maybe the Data Collection section, I suggest introducing the cross-sectional and time-series study designs and relating the measuring the speed and degree of government non-pharmaceutical interventions and outcomes to these two study designs.

14. Methods, Statistical Analysis, “we estimate models that use both time and country fixed effects”: Were there any random effects in the longitudinal models?

15. Methods, Statistical Analysis, “we use the difference in logs as the dependent variable ...”: The outcome must be daily cumulative death events, not daily death events. Otherwise, how did you handle negative difference in the log transformation? What specific longitudinal models were used?

16. Results, Variation in the speed and intensity of government responses, “Figure 1 shows a positive correlation ...”: How did the residuals look like in this OLS regression? Was the OLS regression suitable to examine this relationship? This also applies to Figure 2.

17. Results, Variation in the speed and intensity of government responses: Shouldn’t there be a another model for the relationship between the government SI six weeks prior and the max daily new deaths?

18. Results, Variation in the speed and intensity of government responses, “Figure 4 explores these relationships . . . ”: What was the rationale to select these six countries?

19. Results, Regression Results: Please introduce the cross-country models in the Statistical Analyses section.

20. Figures 1–3: (a) Were all 170 countries included in this analysis? If not, please indicate how many countries were in this analysis and describe the reasons why some countries were not included. (b) What was the criteria for the 12 selected countries? (c) Please remind readers the y-axis was depicted in a log scale.

21. Table S2: Why were there 24 parameters (response indicators), not 9?

Reviewer #2: I apologize for the delay in responding to this review request, which I received on August 13, 2020. Prior to committing to this review, I notified the editor that I could not undertake the review in the prespecified 10-day review period; and that I have two potential competing interests -- namely, that I have met one of the co-authors of the manuscript (D. Sridhar) and am generally inclined to be favorable toward her work, and that (at the time of the review invitation) my own team had an article on this same exact topic being reviewed (Siedner et al. PLoS Med 2020;17:e1003244). In general, I thought that this manuscript was well written and obviously addresses an important public health issue. I also believe that the authors' release of their social distancing database into the public domain was a significant service to the medical/public health community. That said, there are a number of difficult issues related to the cross-sectional analysis and within-country heterogeneity that need to be addressed, however, and until they are, I am unsure as to what can be learned from the analysis. I do not see anything to suggest that this analysis is fatally flawed or that the analysis cannot be suitably revised. I am returning my review on September 11.

Major comments:

Broadly speaking, the authors conduct two sets of analyses: a cross-sectional analysis and a longitudinal analysis.

1. In the cross-sectional analysis, the authors calculate a Stringency Index that is averaged across the entire study period (Jan-May 2020), which is then used to calculate the two explanatory variables of primary interest: average stringency and speed of response. They then correlate these variables with the two outcome variables of primary interest: average daily growth in the death rate and zenith death rate. There are several concerns here:

a) In general, I would like to see more analyses teasing apart the general findings about the Stringency Index. Throughout the Results section, the authors write things like "a 10-point difference in SI would be expected to lead, six weeks later, to a daily growth rate in deaths nearly half a percentage point lower", but what changes would a country government need to implement to do that? eg Can they get there by closing schools and workplaces but keeping bars open?

b) The explanatory and outcome variables of primary interest, insofar as they compress longitudinal information into a single observation per country, mask considerable heterogeneity within countries, both over time (ie., temporal heterogeneity) as well as geographically (ie., regional heterogeneity).

c) It is unclear to me how the authors handled distancing measures that were implemented and then rescinded during the study period. For example, a country may have initiated a "lockdown" in March and then rescinded the lockdown in May. Do the authors use a "once on, always on"/"once treated, always treated" approach here? This is unclear to me from reading the Methods and should be clarified. I _think_ what the authors did is they calculated the Stringency Index for each day in each country, and then averaged across the days. So in a toy example with 3 days, if a country had a Stringency Index value of 0 on day 1, 50 on day 2, and 10 on day 3, the average Stringency Index would be calculated as (0+50+10)/3 = 20. In my understanding, if countries "turned on" and "turned off" social distancing measures at different time points during the study period, the Stringency Index would increase and decrease on a daily basis, but the average Stringency Index over the study period would largely reflect the maximum stringency as well as the number of days that social distancing was "turned on". Some difficulties with this specification can be better understood with a toy example. Supposing for example we have a 10-day study period in which Country A has a Stringency Index of 100 on days 1 & 2, then turns off social distancing completely and has a Stringency Index of 0 on days 3-10. The average Stringency Index for Country A through the study period would therefore be (100+100+0+...+0)/10 = 20. An alternative Country B might have limped along with a Stringency Index of 20 on each day throughout the study period, and would have an average Stringency Index of (20+20+...+20)/10 = 20. Under the authors' scoring, both countries would have equivalent values of the primary explanatory variable despite having arguably very different (formal) social distancing responses. Similarly, compressing the longitudinal data to create the cross-sectional outcome variables (average daily growth in the death rate and zenith death rate) also presents difficulties. Both China and Spain, for example, topped out at around 10,000 daily new cases (I don't know what their comparative death rates were)--and would therefore be treated equivalently in terms of the zenith outcome variable--but China has arguably been much more successful at containing their local epidemic compared with Spain, which is currently in a second surge.

d) For one of the explanatory variables (speed of response), the authors specified this variable as the number of days until the country achieved a Stringency Index of 40. They provide some text to justify this arbitrary threshold, which is appreciated, but I guess I am just not super convinced, partly because it is not even clear to me what a SI=40 means. The authors state that "almost every country reaches at least this threshold at some point"--were there specific components of the Stringency Index (eg., schools, public events) that drove this particular observation? Why not conduct some sensitivity analyses to see whether their findings are sensitive to the threshold choice? Or better yet, why not simply model the slope (ie., "speed of response") directly?

e) In terms of geographic heterogeneity, the authors did not account for the substantial within-country heterogeneity. The US, for example, would treated as a single country with a single observation for level of stringency and outcome. Obviously this is untrue, both in terms of the implementation of social distancing measures (Siedner et al. PLoS Med 2020;17:e1003244) as well as in the relaxation/rescinding of social distancing measures (Tsai et al. medRxiv 2020 Aug 7, doi:10.1101/2020.07.15.20154534)--and not just in the US, but also in other countries with a much stronger national level response, eg., China (Maier & Brockmann, Science 2020;368:742-6; Lai et al. Nature 2020 May 4, doi:10.1038/s41586-020-2293-x). The authors mention this limitation in the Discussion on page 14, but in my opinion this limitation is so glaring that I am not sure what we can learn from the cross-sectional analyses. (Perhaps the authors can be clearer about this in revised text.) The authors cite Cauchemez et al. (ref #40) in noting the significant country-to-country variability in government stringency and the 2009 influenza pandemic. But this is also well known in the context of COVID in the US (eg., Holtz et al. PNAS 2020;117:19837-43). Certainly if the measurement error (ie., due to lack of accounting for within-country heterogeneity) in the outcome variables is non-differential, the estimated associations would simply bias toward the null. But the same cannot be said for measurement error in the explanatory variables. Other investigators have dealt with this by collecting data on social distancing measures implemented at lower/subnational levels of jursdiction (eg., Hsiang et al. Nature 2020;584:262-7) or focused on a single country (as we did in our own work, Siedner et al. PLoS Med 2020;17:e1003244 and Tsai et al. medRxiv 2020 Aug 7, doi:10.1101/2020.07.15.20154534).

2. The authors also conducted longitudinal analyses, which would address concerns about masking within-country temporal heterogeneity (but not concerns about masking within-country geographic heterogeneity). Here they estimate fixed effects regression models, specifying country i on day t as the unit of analysis, lagging the explanatory variable by 6 weeks.

a) The fixed effects estimates are driven entirely by within-country changes in the Stringency Index and by within-country changes in mortality. So I would like to see some descriptive statistics (eg., intraclass correlations etc) quantifying the extent of within vs. between country variation.

b) It is unclear to me why the authors chose a lag of 6 weeks. In my opinion, there is considerable uncertainty about the appropriate interval one would expect between implementation of social distancing measures and any response in mortality. While the median incubation period is likely on the order of 3-5 days, the median time between symptom onset and death varies widely (eg., 8 days in Italy, two weeks in China). (Death probabilities, and time to death, would also be highly contingent on local comorbidity epidemiology as well as health system factors, but these would be unlikely to vary during the short time frame of the study and would therefore be differenced out in the analysis.) In our own work, we took a much more exploratory approach to modeling the association between social distancing and mortality (Siedner et al. PLoS Med 2020;17:e1003244). Perhaps the authors might adopt a similar approach and specify different lag periods to probe the extent to which their findings are sensitive to the choice of lag.

c) The authors motivate their lagged analysis, at the bottom of page 7, by suggesting that reverse causality could be at play. To rule this out, they probably should just probe the data rather than speculate. One could imagine a companion analysis in which the authors estimate the association between lagged deaths and the Stringency Index (to see whether it is indeed the case that "deaths might trigger a policy response"). As above, the extent to which the forward/reverse associations differ would depend on the extent of within-country variation over time.

One more specific point about the analyses:

d) It would be helpful if the authors could write out the estimating equations. It was not clear to me, either from the text or in Table 1, whether the authors estimated a single regression model containing both of the explanatory variables of interest (and also adjusting for the covariates listed in footnotes b & c), or whether the authors estimated two regression models containing 1 of the explanatory variables of interest (each of which adjusted for the covariates listed in footnotes b & c).

Minor comments:

3. The authors justify their use of death rates (rather than case rates) by asserting that they are likely to be reported more consistently. Is this true? (eg., Weinberger et al. JAMA Intern Med 2020 Jul 1, doi:10.1001/jamainternmed.2020.3391; Rivera et al. medRxiv 2020 Jun 27, doi:10.1101/2020.05.04.20090324).

4. Table S1-- Could the authors perhaps be a little more concrete about what goes into the different coding elements? For example, are restaurants included in C2 (workplaces) or C4 (restrictions on gatherings)? Are churches included in C3 (public events) or C4 (gatherings)?

5. To calculate the composite Stringency Index, the authors simply added up the individual components listed in Table S1. Obviously there are problems with granting equal weights to the different components, eg. canceling all public events (2 points) vs. closing some schools (also 2 points). This should be mentioned as a limitation.

b) To avoid imposing equal weights on the different elements, why not extract the first principal component (Table S2) and use that as the Stringency Index? Such an approach would allow the data to drive the weighting, similar to Filmer & Pritchett (Demography 2001;38:115-132).

6. While the Discussion section refers vaguely to "potential confounders", there is no mention of spontaneous behavioral change that may have occurred prior to implementation of any distancing measures. Such behavioral change was highly conditioned, both at the individual level (eg, Papageorge et al. COVID Econ 2020;40:1-45; Weill et al PNAS 2020 Aug 18;117(33):19658-19660) and at the area level (eg., Wright et al. SSRN 17 Jun 2020, doi:10.2139/ssrn.3573637), by preexisting sociodemographic factors, which is important from an equity perspective; and this evidence of pre-policy behavioral change also reinforces the notion that the epidemic itself, rather than lockdown-style distancing measures, was the cause of any observed economic fallout (eg., Sheridan et al. PNAS 2020 Aug 25;117 (34):20468-20473). To the extent that implementation of any distancing measures may have been implemented in response to either a worsening local epidemic or behavior change already occurring, the authors estimates are likely to be biased away from the null rather than toward the null--which would lead them to overestimate the effectiveness of social distancing measures in curbing the COVID epidemic.

7. There are several priority statements scattered throughout the manuscript (eg, page 3, "most comprehensive view"; page 15, "most comprehensive test"; etc). I am uncertain as to whether this is truly the case. For example, Islam et al. (BMJ 2020;370:m2743-- not cited in this manuscript, which is probably an oversight) used the authors' data on distancing measures to estimate the association between distancing measures and COVID-19 incidence-- this does not necessarily undercut the novelty of the database on distancing measures (although cf. Cheng et al. Nature Hum Behav 2020;4:756-768; Zheng et al. Scientific Data 2020;7:286; and Desvars-Larrive et al. Scientific Data 2020;7:285-- I am sure there are others), but it does tend to undercut the novelty of the analyses presented in this specific manuscript. In any case, the manuscript would probably be improved with (a) more comprehensive referencing of these alternative efforts; (b) better contextualizing of the present analyses in relation to other published literature and preprints (how does the authors' database compare with these other efforts? what are the gaps in the alternative databases and how do they compare with the authors' database? etc); and (c) deleting the priority claims, which are probably unnecessary anyway.

Alexander Tsai, MD

Massachusetts General Hospital

6. PLOS authors have the option to publish the peer review history of their article (what does this mean?). If published, this will include your full peer review and any attached files.

Reviewer #1: No

Reviewer #2: **Yes: **Alexander Tsai

---

## [Author Response · Author response to Decision Letter 0]

13 May 2021

We thank the reviewers for their careful reading of the manuscript, and have endeavoured to respond to their helpful suggestions. Please see the attached file for detailed responses.

---

## [Decision Letter · Decision Letter 1]

31 May 2021

Government responses and COVID-19 deaths: global evidence across multiple pandemic waves

PONE-D-20-20672R1

Dear Dr. Hale,

We’re pleased to inform you that your manuscript has been judged scientifically suitable for publication and will be formally accepted for publication once it meets all outstanding technical requirements.

Kind regards,

Holly Seale

Academic Editor

PLOS ONE

Additional Editor Comments (optional):

Reviewers' comments:

Reviewer's Responses to Questions

**Comments to the Author**

1. If the authors have adequately addressed your comments raised in a previous round of review and you feel that this manuscript is now acceptable for publication, you may indicate that here to bypass the “Comments to the Author” section, enter your conflict of interest statement in the “Confidential to Editor” section, and submit your "Accept" recommendation.

Reviewer #2: All comments have been addressed

2. Is the manuscript technically sound, and do the data support the conclusions?

Reviewer #2: Yes

3. Has the statistical analysis been performed appropriately and rigorously? 

Reviewer #2: Yes

4. Have the authors made all data underlying the findings in their manuscript fully available?

Reviewer #2: Yes

5. Is the manuscript presented in an intelligible fashion and written in standard English?

Reviewer #2: Yes

6. Review Comments to the Author

Reviewer #2: I am identified as reviewer #2 on the initial submission. The authors have provided thoughtful replies to the comments that I raised. (Many of the comments I initially raised have been dealt with either with the deletion of the cross-sectional analysis or with the publication of their Nature Human Behavior data paper.)

7. PLOS authors have the option to publish the peer review history of their article (what does this mean?). If published, this will include your full peer review and any attached files.

Reviewer #2: **Yes: **Alexander Tsai, MD

---

## [Editor Report · Acceptance letter]

30 Jun 2021

PONE-D-20-20672R1 

Government responses and COVID-19 deaths: global evidence across multiple pandemic waves. 

Dear Dr. Hale:

I'm pleased to inform you that your manuscript has been deemed suitable for publication in PLOS ONE. Congratulations! Your manuscript is now with our production department. 

Kind regards, 

on behalf of

Dr. Holly Seale 

Academic Editor

PLOS ONE